# The impact of a named GP scheme on continuity of care and emergency hospital admission: a cohort study among older patients in England, 2012–2016

Peter Tammes,  Rupert A Payne, Chris Salisbury, Melanie Chalder, Sarah Purdy, Richard W Morris

Bristol Medical School: Population Health Sciences, Centre for Academic Primary Care, University of Bristol, Bristol, UK

**Correspondence to**
Dr Peter Tammes;
p.tammes@bristol.ac.uk

## ABSTRACT

**Objective** To investigate whether the introduction of a named general practitioner (GP, family physician) improved patients' healthcare for patients aged 75 and over in England.

**Setting** Random sample of 27 500 patients aged 65 to 84 in 2012 within 139 English practices from the Clinical Practice Research Datalink linked with Hospital Episode Statistics.

**Design** Prospective cohort approach, measuring patients' GP consultations and emergency hospital admissions 2 years before/after the intervention. Patients were grouped in (i) aged over 74 and (ii) younger than 75 in both periods in order to compare who were or were not subject to the intervention. Adjusted associations between the named GP scheme, continuity of care and emergency hospital admission were examined using multilevel modelling.

**Intervention** National Health Service policy to introduce a named accountable GP for patients aged over 74 in April 2014.

**Main outcome measures** (A) Continuity of care index-score, (B) risk of emergency hospital admissions, (C) number of emergency hospital admissions.

**Results** The intervention was associated with a decrease in continuity index-scores of −0.024 (95% CI −0.030 to −0.018, p<0.001); there were no differences in the decrease between the two age groups (−0.005, 95% CI −0.014 to 0.005). In the pre-intervention and post-intervention periods, respectively, 15.4% and 19.4% patients had an emergency admission. The probability of an emergency hospital admission increased after the intervention (OR 1.156, 95% CI 1.064 to 1.257, p=0.001); this increase was bigger for patients over 74 (relative OR 1.191, 95% CI 1.066 to 1.330, p=0.002). The average number of emergency hospital admissions increased after the intervention (rate ratio (RR) 1.178, 95% CI 1.103 to 1.259, p<0.001); this increase was greater for patients over 74 (relative RR 1.143, 95% CI 1.052 to 1.242, p=0.001).

**Conclusion** The introduction of the named GP scheme was not associated with improvements in either continuity of care or rates of unplanned hospitalisation.

### Strengths and limitations of this study

► This study was the first to investigate the relationship between the introduction of a named general practitioner (GP) scheme and continuity of primary care and unplanned hospitalisation.
► This study took a 4 year observational period into account, namely 2 year periods before and after the implementation of the named GP policy.
► This cohort study used individual electronic healthcare records data from the Clinical Practice Research Datalink linked with Hospital Episode Statistics.
► Our data set did not make it possible to specify the named GP assigned to a patient.
► As all older patients included in the sample survived the 4 year observational period, this might reduce the generalisability of our findings.

## INTRODUCTION

Nearly every country in the world is experiencing growth in the number and proportion of older persons.[1] It is projected that by 2019 one in eleven people in the UK will be aged 75 or older, increasing to one in seven by 2040.[2] This demographic trend is likely to increase the number and proportion of people with long-term disability and chronic or multiple health conditions. To meet the challenges of an ageing population and to better serve those living with complex health and care needs, in April 2014 the National Health Service (NHS) Employers and General Medical Services agreed to introduce a named accountable general practitioner (GP) (family or primary care physician) for all patients aged 75 or more. The aim was to provide personalised, proactive care to keep older people healthy, independent and out of hospital.[3 4]

BMJ

In the UK, patients are registered at one general (family) practice but might see different physicians within that practice. The introduction of a named GP scheme was intended to facilitate older patients consulting the same doctor, thus improving continuity of primary care (ie, seeing the same clinician over time). As there is some evidence that continuity of primary care is declining in England,[5] the named GP scheme could potentially reverse this trend.

A key objective of the introduction of the named GP scheme was to avoid or decrease hospital admission. Previous systematic reviews based on international literature concluded that better clinician continuity of care reduces hospitalisation.[6 7] There is some evidence from the UK that patients who do not see the same GP over a period of time are at higher risk of emergency hospital admission and have more admissions than those who see the same or a small number of GPs.[8 9] The aim of the current study was to investigate whether the named GP scheme improved older patients' continuity of primary care, and decreased older patients' risk of emergency hospital admission and the number of such admissions. The findings of this study might also be relevant for other (Western) countries as they face the same burdens from an ageing population on their healthcare systems.

## METHODS
### Study design and setting
This study used data from the Clinical Practice Research Datalink (CPRD) which contains anonymised electronic healthcare records on 4.4 million patients (6.9% of the UK population) and is nationally representative in terms of age, sex and ethnicity.[10] We obtained a random sample of 27 500 patients in 139 English GP practices (family practices) who were aged 65 to 84 in 2012, alive in 2016, registered with their GP practice for at least 1 year prior to 2012, and not transferred out of the GP practice during the study's observation period. The CPRD was deterministically linked with Hospital Episode Statistics to identify emergency hospital admissions in the financial years April 2012 to March 2016. This allowed us to compare a patient's individual healthcare use 2 years before the introduction of the named GP scheme (April 2012 to March 2014) with their healthcare use 2 years after its introduction (April 2014 to March 2016). To measure the impact of the introduction of a named GP scheme appropriately, we excluded patients who became 75 years old during the study (n=5703) and created two age groups for patients who were aged 75 years and over in both periods (n=9682), or aged under 75 years in both periods (n=12 115) in order to compare groups who were or were not subject to the intervention. Furthermore, analyses were restricted to only those patients with at least two GP consultations allowing to calculate continuity of primary care scores in either the pre-intervention or post-intervention periods, totalling 19 235 patients in the

pre-intervention period and 19 265 patients in the post-intervention period.

Based on our previous work, we expected 13.5% to undergo an emergency admission over a single year, therefore over 2 years, we estimate 22% will experience an admission.[9] The results of that study suggested that an upward shift across a quartile of the distribution of continuity of care might decrease risk by approximately 10%. Comparing a subsequent 2-year period in which the rate is reduced by 10% (ie, rate ratio of 0.9) from 22% to 19.8% by introduction of the named GP scheme for patients aged 75 and over, we had over 99% power to detect this difference at the 5% significance level at the given sample size for the pre-intervention and post-intervention periods. If the rate is reduced by 6% (instead of 10%) to 20.7%, we had 88% power to detect this difference. These calculations were done in Stata 15.1 using sampsi and power commands.

### Exposure
After the introduction of the named GP scheme in April 2014, all patients aged 75 and over had to be notified by letter or during a consultation by 30 June 2014 of their named and accountable GP, or within 21 days after a patient became 75 or was newly registered if their practice was not already operating a personal list.[3] Practices were required to enter the Read Code 'Informing patient of named accountable general practitioner' (code 67DJ) in the patient's health record. Based on the date of recording of this Read Code for all the 12 526 patients 75 and older in 2014 in the data set, 65% were informed before 1 July 2014 and 75% before 1 August 2014. At the end of the observation period, that is, 1 April 2016, 97% had been notified (see online supplementary figure 1). We did not, however, exclude patients who were notified after 30 June 2014 for the purposes of our study as we did not know which practices were already operating a personal list. Furthermore, our analysis focused on the effects of the policy intervention as it was implemented; in effect, an intention-to-treat analysis.

### Outcome measures
Three outcome measures were assessed, the first being change in continuity of primary care. We used a combination of CPRD staff codes to identify GP staff (senior partners, partners, salaried doctors, locum doctors and GP registrars) within the practice and dates of consultations to identify whether these occurred during the pre-intervention or post-intervention period. Consultations included clinic and surgery consultations, home visits, out-of-hours' visits, telephone consultations and third-party consultations. Following Hobbs et al[11] we did not restrict according to consultation duration. Where a patient had more than one contact per day, we used information about staffing relating to the first consultation only, to avoid potential concerns about duplication of consultations. This information allows the quantification of continuity of care over 2 year periods pre-intervention

and post-intervention. We calculated continuity of care using the Bice & Boxerman (BB) index,[12] which ranges between 0 (complete absence of continuity) and 1 (perfect continuity of care), as this has been recommended for use in primary care research.[13] BB index-scores can be calculated for patients who consulted a GP more than once.

The second outcome assessed was change in probability of experiencing at least one emergency hospital admission after the second GP consultation in both the pre-intervention and post-intervention periods. We made no distinction between admissions by specific routes (eg, patient self-presentation to the emergency department, GP referral to a hospital speciality). A patient's probability of at least one emergency hospital admission may not necessarily reflect the number of admissions a patient experienced. The probability might have decreased while the average number of emergency hospital admissions might have increased and vice versa. The third outcome, therefore, was the number of emergency hospital admissions after the second GP consultation in both the pre-intervention and post-intervention periods, categorised as zero, one, two and three or more.

## Covariates

Our choice of covariates was guided by the QAdmission score,[14] previously developed using data from a similar routine general practitioner database to predict hospital admissions. Complete data were available on all patients in the sample with regard to age, gender, number of GP consultations, area-based socioeconomic Index of Multiple Deprivation 2015 in quintiles, location (conurbation, urban, rural) and the following morbidities measured between April 2010 and March 2014 for the pre-intervention period and measured between April 2012 and March 2016 for the post-intervention period. These included diagnoses made in the 2 years prior to the pre-intervention and post-intervention periods, using published clinical code lists as collected in the Manchester Clinical Codes repository[15]: epilepsy,[16] chronic kidney disease,[17] cancer,[18] asthma,[17] stroke,[19] coronary heart disease,[19] diabetes,[19] chronic obstructive pulmonary disease,[16] depression[20] and schizophrenia.[20] Furthermore, we took into account clustering at the practice level as practice factors might facilitate or reduce continuity of care[21] and estimated the number of GPs in a practice using patient's GP consultations and staff role information for each general practice. Descriptive statistics are provided in table 1.

To adjust for continuity of care (BB index-score) for the second outcome, we determined the continuity of care until an emergency hospital admission or to the end of each period (whichever came first) when not having experienced such an emergency admission, excluding from the analysis patients who experienced an emergency admission before their first or second GP consultation (as these patients would not have a continuity score). This resulted in a reduction in the of number of observations

from 38 500 to 37 207. The BB index-scores were divided into quintiles.

## Statistical methods

A patient's BB index and a patient's emergency hospital admission were measured for both the pre-intervention and the post-intervention periods. To account for repeated measurements by time, by patient and by practice, this study used multilevel modelling. Because continuity of care was a continuous variable, a normal response regression model was used to associate the named GP scheme with continuity of care (BB index-score). Because experiencing at least one emergency hospital admission was a binary variable, a binomial logit regression model to associate the named GP scheme with risk of emergency hospital admissions was used. A Poisson regression model was used to associate the named GP scheme with the number of emergency hospital admissions as this outcome was a count.

To represent whether the effect of the intervention operated differently for patients aged over 75 (exposed) from those aged under 75 (unexposed), we included the age × period interaction. This could be interpreted as difference in change of the BB index score, the relative OR of emergency hospital admission or the relative rate ratio of number of emergency hospital admissions. We used Stata 15.1 to perform our analyses.

## RESULTS
### Outcome: continuity of primary care

The distribution of the BB index varied widely, with similarity between the pre-intervention and post-intervention distributions (figure 1). Respectively, 1365 (7.1%) and 2523 (13.1%) patients never or always saw the same GP in the pre-intervention period: equivalent numbers were 1376 (7.1%) and 2086 (10.8%) in the post-intervention period. The change in BB index-score over time also varied widely (see online supplementary figure 2) with an IQR between −0.190 and 0.141.

The BB index-score decreased over time by 0.028 (from 0.428 in the pre-intervention period to 0.399 in the post-intervention); this equates to a drop in the mean continuity of care by about 6.5%. The BB index-score for patients aged 75 and over decreased from 0.434 pre-intervention to 0.403 post-intervention (a mean decrease of 0.031). This decrease was slightly bigger than for patients younger than 75, from 0.422 pre-intervention to 0.397 post-intervention (a drop of 0.025). An unadjusted multilevel (normal response) model for continuity of care (BB index-score) showed there was no evidence that this decrease in continuity of care following the intervention differed between the two age groups (table 2, time-age interaction −0.006 (95% CI −0.015 to 0.004)). As patients in the lowest or highest continuity of care quartiles consulted a GP less often than those in the middle two quartiles

**Table 1** Descriptive statistics

| | Pre-intervention (19 235) | Post-intervention (19 265) |
|---|---|---|
| | N (Pct.) | N (Pct.) |
| Patients younger than 75 | 10 404 (54.1) | 10 368 (53.8) |
| Patients 75 or older | 8831 (45.9) | 8897 (46.2) |
| Male | 8699 (45.2) | 8698 (45.2) |
| Female | 10 536 (54.8) | 10 567 (54.8) |
| Least deprived, quintile 1 | 5294 (27.5) | 5340 (27.7) |
| Deprivation quintile 2 | 4395 (22.9) | 4421 (23.0) |
| Deprivation quintile 3 | 4266 (22.6) | 4238 (22.0) |
| Deprivation quintile 4 | 3195 (16.6) | 3194 (16.6) |
| Most deprived, quintile 5 | 2084 (10.8) | 2071 (10.7) |
| 2–5 GP consultations (2-5), quintile 1 | 5333 (27.7) | 5130 (26.6) |
| 6–9 GP consultations, quintile 2 | 4697 (24.4) | 4468 (23.2) |
| 10–15 GP consultations, quintile 3 | 4545 (23.6) | 4518 (23.5) |
| 16 or more GP consultations (16+), quintile 4 | 4660 (24.2) | 5149 (26.7) |
| Less than 9 GPs in practice pre-intervention (post:<8), quintile 1 | 2977 (15.5) | 2764 (14.4) |
| 9–14 GPs in practice pre-intervention (post: 8–13), quintile 2 | 4715 (24.5) | 4555 (23.6) |
| 15–21 GPs in practice pre-intervention (post: 14–22), quintile 3 | 5032 (26.2) | 5996 (31.1) |
| More than 21 GPs in practice pre-intervention (post >22), quintile 4 | 6511 (33.9) | 5950 (30.9) |
| Urban conurbation | 6145 (32.0) | 6180 (32.1) |
| Cities and towns | 10 207 (53.1) | 10 290 (53.4) |
| Rural | 2883 (15.0) | 2795 (14.5) |
| No emergency hospital admission | 16 269 (84.6) | 15 520 (80.6)) |
| One emergency hospital admission | 2070 (10.8) | 2368 (12.3) |
| Two emergency hospital admissions | 557 (2.9) | 750 (3.9) |
| More than two emergency hospital admissions | 339 (1.7) | 627 (3.2) |

GP, general practitioner.

| | Median (IQ) | Median (IQ) |
|---|---|---|
| Total number of morbidities* (0–6) | 1 (0–1) | 1 (0–1) |
| Continuity of care (BB index-score) patient-level (0–1) | 0.344 (0.184–0.622) | 0.333 (0.167–0.574) |
| Continuity of care (BB index-score) practice-level (0–1) | 0.416 (0.321–0.541) | 0.397 (0.306–0.517) |

*Diagnosed with one or more of the following seven chronic conditions: chronic renal disease, cancer, asthma, stroke, coronary heart disease, diabetes or COPD.
BB, Bice & Boxerman; COPD, chronic obstructive pulmonary disease; IMD, index of multiple deprivation; Q, quartiles.

(see online supplementary table 1), we included number of consultations in the analysis as one of the covariates together with other factors such as gender, number of chronic conditions, socioeconomic deprivation, number of GPs in practice and rurality (see online supplementary table 2). This adjusted model still showed no significant difference in the decrease of continuity of care over time between the two age groups (table 2, time-age interaction −0.005 (95%CI −0.014 to 0.005)). Continuity of care declined in both the unexposed and exposed groups and there was no evidence of the decline being stronger in one of the groups.

**Sensitivity analysis**

We also calculated for each practice the average practice-level continuity of care score over 2012 to 2014, having divided practices into tertiles: low, middle, high continuity of care. This allowed us to determine whether patients in practices with different levels of continuity of care show differing trends in continuity of care post-intervention. The result of an interaction between period, age and practice-level continuity of care is illustrated in figure 2. The continuity of care of patients in a practice with generally low continuity of care dropped less between pre-introduction and post-introduction of the named GP scheme for both patients younger and older

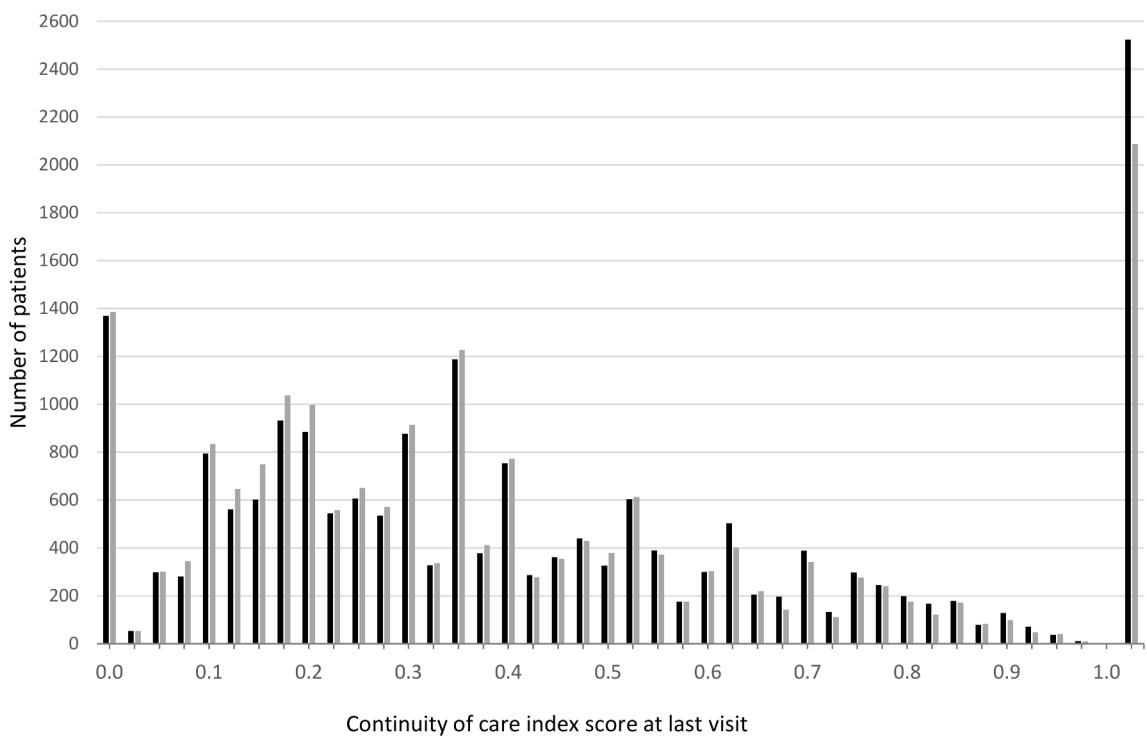

**Figure 1** Distribution of Bice & Boxerman index-scores for April 2012 to March 2014 (19 235 patients) and for April 2014 to March 2016 (19 265 patients).

than 75 compared with practices with generally middle and high practice-level continuity of care.

### Outcome: risk of emergency hospital admission

In the pre-intervention and the post-intervention periods, respectively, 2966 (15.4%) and 3745 (19.4%) patients had one or more emergency admissions. The probability of an emergency hospital admission for patients aged 75 and over showed an absolute increase of 6.3% points over time (from 19.9% pre-intervention to 26.2% post-intervention). There was evidence from the unadjusted model that the relative increase in odds of admission was 17.9% (95% CI 5.9% to 31.4%) greater in those aged over 75 years than those aged under 75 years after the introduction of the named GP scheme (table 3). This relative difference between

age groups persisted (19.1%, 95% CI 6.6% to 33.0%) after adjustment for other factors such as gender, number of chronic conditions, socioeconomic deprivation, number of GPs in practice and rurality (see online supplementary table 3). The relative difference between age groups was marginally greater following additional adjustment for continuity of care (BB index-score estimated until the event date) and number of GP consultations (22.8%, 95% CI 8.6% to 38.8%).

### Outcome: number of emergency hospital admissions

In the pre-intervention period 16 269 (84.6%), 2070 (10.7%), 557 (2.9%) and 339 (1.76%) experienced respectively no, one, two or three or more emergency hospital admissions. In the post-intervention period

**Table 2** Estimates of B-coefficients from multilevel regression (normal response) model for the association between introduction of named GP and continuity of care (Bice & Boxerman index-score), England 2012 to 2016 (38 500 observations)

| | Unadjusted model | | | Adjusted model* | | |
|---|---|---|---|---|---|---|
| | Coef. | P value | 95% CI | Coef. | P value | 95% CI |
| Constant | 0.440 | <0.001 | 0.413 to 0.467 | 0.427 | <0.001 | 0.404 to 0.449 |
| Period (ref.=pre) | −0.024 | <0.001 | −0.031 to −0.018 | −0.024 | <0.001 | −0.030 to −0.017 |
| Age (ref.=<75) | 0.013 | 0.001 | 0.005 to 0.021 | 0.017 | <0.001 | 0.009 to 0.025 |
| Period* Age | −0.006 | 0.240 | −0.015 to 0.004 | −0.005 | 0.342 | −0.014 to 0.005 |

*Co-variates set to average: gender, number of chronic conditions, level of deprivation (quintiles), number of GPs in practice (quintiles), number of GP consultations (quartiles) and urban/rural practice location. For complete table, see online supplementary table 2.
GP, general practitioner.

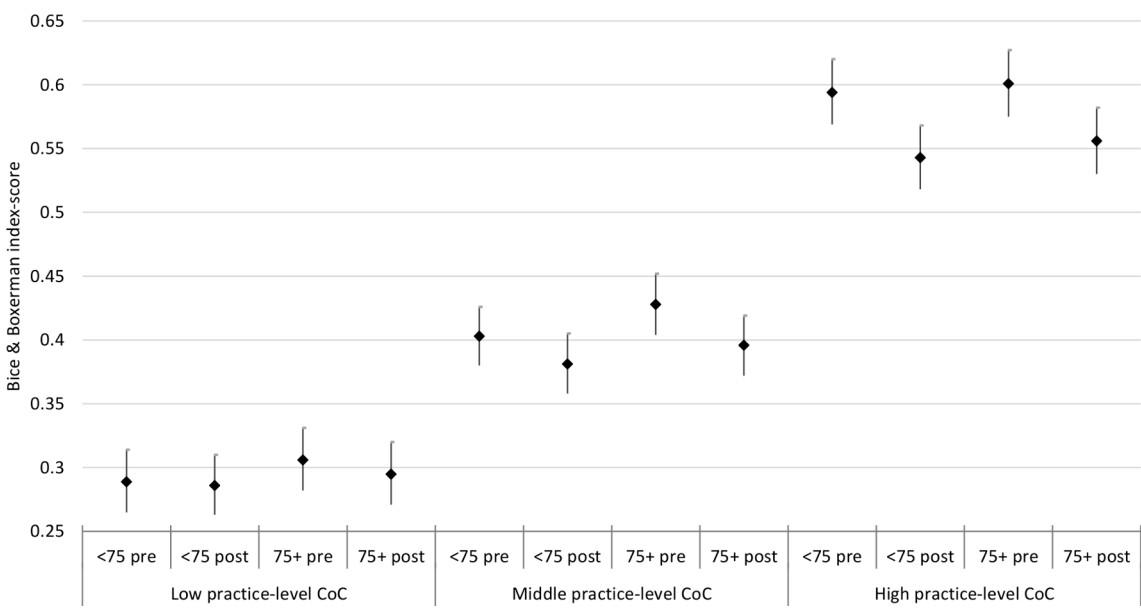

**Figure 2** Estimates of B-coefficients (95% CIs) from multilevel regression (normal response) model for the association between introduction of named GP and continuity of care (Bice & Boxerman index-score), split according to level of practice-level continuity of care. CoC, continuity of care; GP, general practitioner.

15 520 (80.6%), 2368 (12.3%), 750 (3.9%) and 627 (3.3%) experienced respectively no, one, two or three or more emergency hospital admissions. The mean number of emergency hospital admissions for patients aged 75 and over showed an absolute increase of 0.154 over time (from 0.313 pre-intervention to 0.467 post-intervention). There was evidence from the unadjusted model that the relative increase in mean number of emergency hospital admissions after the introduction of the named GP scheme was 14.6% (95% CI 5.5% to 24.5%) greater in those aged over 75 years than those aged under 75 years (table 4, Unadjusted model). This relative difference between age groups persisted (14.3%, 95% CI 5.2% to 24.2%) after adjustment for other factors such as gender, number of chronic conditions, socioeconomic deprivation, number of GPs in practice and rurality (table 4, Adjusted model 1; see online supplementary table 4).

### Sensitivity analysis

To adjust for continuity of care (BB index-score) and for number of GP consultations we determined the continuity of care at the end of each period for all patients included in the analysis, in contrast to the other outcome where continuity of care was estimated only until the event date. In this model the relative difference between the two age groups was slightly lower than in the unadjusted model (12%, 95% CI 3.1% to 21.5%) (table 4, Adjusted model 2).

### DISCUSSION
### Principal findings

Continuity of care decreased between the pre-intervention and post-intervention periods and this decrease was similar for patients aged between 65 and 74 (who were not eligible for the named GP scheme over the period of

**Table 3** Estimates of odds ratios (OR) from multilevel regression (binomial logit) model for the association between introduction of named GP and risk of an emergency hospital admission, England 2012 to 2016 (38 500 observations)

|  | Unadjusted model | | | Adjusted model 1* | | | Adjusted model 2† | | |
|---|---|---|---|---|---|---|---|---|---|
|  | OR | P value | 95% CI | OR | P value | 95% CI | OR | P value | 95% CI |
| Period (ref.=pre) | 1.206 | <0.001 | 1.111 to 1.309 | 1.156 | 0.001 | 1.064 to 1.257 | 1.137 | 0.007 | 1.035 to 1.254 |
| Age (ref.=<75) | 1.887 | <0.001 | 1.736 to 2.048 | 1.594 | <0.001 | 1.464 to 1.735 | 1.680 | <0.001 | 1.530 to 1.846 |
| Period* Age | 1.179 | 0.003 | 1.059 to 1.314 | 1.191 | 0.002 | 1.066 to 1.330 | 1.228 | 0.001 | 1.086 to 1.388 |

*Covariates set to average: gender, number of chronic conditions, level of deprivation (quintiles), number of GPs in practice (quartiles) and urban/rural practice location. For complete table with B-coefficients, see online supplementary table 3.
†Included also standardised covariates: number of GP consultations (quartiles) and continuity of care (Bice & Boxerman index-score, quartiles) till first emergency hospital admission or the end of the observation period when not admitted. Number of observations: 37 207.
GP, general practitioner.

**Table 4** Estimates of rate ratios (RR) from multi-level regression (Poisson) model for the association between introduction of named GP and the number of emergency hospital admissions, England 2012 to 2016 (38 500 observations)

| | Unadjusted model | | | Adjusted model 1* | | | Adjusted model 2 | | |
|---|---|---|---|---|---|---|---|---|---|
| | RR | P value | 95% CI | RR | P value | 95% CI | RR | P value | 95% CI |
| Period (ref.=pre) | 1.249 | <0.001 | 1.170 to 1.332 | 1.178 | <0.001 | 1.103 to 1.259 | 1.171 | <0.001 | 1.097 to 1.250 |
| Age (ref.=75-) | 1.821 | <0.001 | 1.687 to 1956 | 1.571 | <0.001 | 1.462 to 1.690 | 1.372 | <0.001 | 1.280 to 1.470 |
| Period* Age | 1.146 | 0.001 | 1.055 to 1.245 | 1.143 | 0.001 | 1.052 to 1.242 | 1.120 | 0.007 | 1.031 to 1.215 |

*Covariates set to average: gender, number of chronic conditions, level of deprivation (quintiles), number of GPs in practice (quartiles) and urban/rural practice location. For complete table, see online supplementary table 4.
†Included also standardised covariates: number of GP consultations (quartiles) and continuity of care (Bice & Boxerman index-score, quartiles).
GP, general practitioner.

study) and patients aged 75 and over (who were eligible). Over time, continuity of care for patients aged 75 years or over declined less in practices which had lowest continuity of care at baseline. The average decrease in continuity of care was small, about 6.5% from baseline, although there was considerable variation across patients and practices. The probability of an emergency hospital admission increased between the pre-intervention and post-intervention periods and this increase was greater for patients 75 and over. The average number of emergency hospital admissions also increased, and this increase was greater for patients aged 75 and over. In general, the introduction of a named GP scheme was not associated with improvements in either continuity of care or rates of emergency hospital admissions.

Emergency hospital admission showed a stronger increase among patients aged 75 and over, contrary to what we expected, but we don't think that this is associated to the measured decrease in continuity of care as patients 75 and over and those younger than 75 experienced a similar drop in continuity. It is unclear whether the increase is due to the named GP system mediated through some other mechanism than continuity of care, or whether it's due to other factors not captured by our study.

### Strengths and weaknesses of the study

This study used longitudinal individual-level data from older patients in the CPRD to assess continuity of care and its relationship with the incidence of unplanned hospital admission before and after the introduction of a named GP scheme, by comparing patients assigned a named GP with slightly younger patients not assigned a named GP. This allowed us to determine and compare continuity of care and unplanned hospital admission over time and between the affected and unaffected group. The observation period was a 2-year period before and after the introduction of a named GP scheme, allowing us to calculate robust continuity of care scores for each period using the BB index-scores. The data set allowed us also to control for practice-level, clinical and demographic covariates.

The study had some limitations. First, all the patients included in the sample survived the 4-year observational period. This may indicate that we had a particularly 'healthy' group of older patients and might, therefore, reduce the generalisability of the study's findings. Our data set did not make it possible to specify the named GP assigned to a patient, therefore we could not use other measures of continuity of care such as the provider identification index.[22]

### Comparison with other studies

Lloyd and Steventon published a protocol for a regression discontinuity study to investigate the effect of the introduction of the named GP scheme on the number of GP contacts per patients, the number of GP referrals to specialists and the number of common diagnostic tests.[23] Following up on their protocol, Barker, Lloyd and Steventon did not find any associations with their outcomes of interest measured over 9 months following assignment to a named accountable GP and attributed this to their short period under study.[24] The present study took a longer observational period into account, namely 2-year periods before and after the implementation, resulting in more robust findings. Whereas the study of Barker, Lloyd and Steventon focused on number of GP contacts, our study explored the possible effect on continuity of care, since one of the mechanisms by which assigning a named GP to a patient could have an impact might be by increasing continuity. Furthermore, Lloyd and Steventon's outcome measures reflect only primary care service use. As a key objective of the introduction of a named GP scheme was to avoid or decrease hospital admission, this study also calculated associations between the introduction of a named GP and risk and number of emergency hospital admissions.

Using aggregated practice-level data from the GP patient survey, Levene *et al* showed that the proportion of patients seeing their preferred GP dropped between 2012 and 2017, especially in practices with higher percentages of those aged 75 and older.[25] Based on this result they questioned the effectiveness of the named GP policy for older patients. Our study showed a decrease as

well in continuity of care, measured by BB index-scores. However, as our study used individual-data, avoiding the ecological fallacy, we showed that the decrease in continuity of care was similar for those aged between 65 to 74 and for those aged 75 and older. Possibly because most patients were already listed at a GP list and introducing a named GP policy for older patients might not have changed their situation of being allocated to a GP much. We were also able to determine that continuity of care of patients in a practice providing on average low continuity of care dropped less compared with patients in practices providing on average high continuity of care, which may be an example of regression to the mean.

### Meaning of the study: possible explanations and implications for clinicians and policymakers

The named GP scheme for older patients was introduced by the NHS, with each individual general practice having to assign doctors to older patients on their list. The General Medical Services Contract did not advise practices to consult patients about their preferred GP as part of this assignment process, nor did it guarantee that patients would see the same clinician at each consultation. However, where patients expressed a preference as to which GP they have been assigned, the practice had to make reasonable effort to accommodate these requests.[3] In most general practices in the UK patients were already nominally allocated to a particular GP within a practice on the practice computer system, because until 2004 patients were registered with an individual GP rather than a practice. However, patients may not have been aware of this, and the GP named on the computer system may have had little or no significance for patient care.[26] The main change introduced with the named GP policy was informing patients of the GP who was accountable for their care. This did not necessarily reflect which GP the patient had seen most often or take into account whether the patient had a preferred GP. Even though the impact might therefore have been expected to be small, this study still provides insights into whether or not this policy has impacted on continuity of care, as well as whether it has achieved its aims of reducing hospitalisation.

Allocating a GP does not imply that patients are able to see or speak to that GP whenever they require advice or care since this depends on GP workload, practice opening hours, salaried and part-time working contracts.[27] The importance of continuity of care in the patient-doctor relationship is much more complex than the simple allocation of a named doctor. Other factors that may be important, particularly in the context of reducing future emergency admissions, are the education of patients over a period of time, and knowledge of a patient's usual health status.[28] These are reflections of the depth of the relationship between the patient and doctor – which will not automatically be improved by the allocation of a named doctor to a particular patient.[29]

A policy of allocating a named GP in itself is not effective and more sophisticated interventions would be needed to improve continuity of care in the UK or countries with similar healthcare systems. However, it is not possible to tell from our study whether applying an assigned named GP scheme in a country where continuity of care is not common, might actually lead to improvements in continuity of care and, consequently, hospital use.

### Unanswered questions and future research

Future research might focus on differences between practices concerning the implementation of the named GP scheme. As this research showed a difference between patients in general practices with on average low versus high continuity of care, a number of other differences could impact implementation, such as practice size and proportion of part-time GPs. Our study focused on continuity of care and unplanned hospital admission. Future research using a 2year or even longer observational period might focus on other healthcare use such as number of GP referrals or diagnostic tests,[24] drug prescription and medication adherence. A complication, however, might be the introduction of a named GP scheme for all patients in April 2015 which should have been implemented in all practices before April 2016.[30] However, the named GP for patients younger than 75 has largely a role of oversight for a patient's health in contrast to the named GP for patients 75 and over who should actively provide personalised care.

This study does not investigate the views and experiences of patients or practice staff. Evidence suggests that older patients value continuity of care,[27] but qualitative research or surveys could explore whether they identified any change in care after the introduction of the named GP scheme. We also do not know whether the scheme led to any meaningful changes in how practices offered care to patients and or in the extent to which individual GPs felt accountable for particular patients. Qualitative research in practices could usefully explore this issue in order to improve implementation of a named GP scheme.

**Contributors** PT and RM designed the study. PT managed and performed the analysis. CS, MC, RAP and SP contributed to the methodological approach and also added significant input to the results and discussion. All authors contributed to the interpretation of the findings. PT was the lead investigator for the overall project.

**Funding** This work was funded by the National Institute for Health Research School of Primary Care Research (NIHR SPCR) grant funded round 13, PI PT project number 337. The views expressed are those of the authors and not necessarily those of the NIHR, NHS or Department of Health. This study is based in part on data from the Clinical Practice Research Datalink obtained under licence from the UK Medicines and Healthcare products Regulatory Agency. The data is provided by patients and collected by the NHS as part of their care and support. The interpretation and conclusions contained in this study are those of the author/s alone.

**Competing interests** None declared.

**Patient consent for publication** Not required.

**Ethics approval** This study was approved by the CPRD Independent Scientific Advisory Committee (ISAC) committee. Protocol no.: 17_140R.

**Provenance and peer review** Not commissioned; externally peer reviewed.

**Data availability statement** Data may be obtained from a third party and are not publicly available.

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
