## [Reviewer comments · BMJ Open]

ARTICLE DETAILS

TITLE (PROVISIONAL)	The impact of a named GP-scheme on continuity of care and emergency hospital admission. A cohort study among older patients in England, 2012-2016.
AUTHORS	Tammes, Peter; Payne, Rupert; Salisbury, Chris; Chalder, Melanie; Purdy, Sarah; Morris, Richard

VERSION 1 – REVIEW

REVIEWER	Marijke Olthof Dept. Family Medicine University Medical Centre Groningen, the Netherlands
REVIEW RETURNED	21-Jan-2019

GENERAL COMMENTS	This study has analyzed the impact of a named GP-scheme in het UK on the continuity of care and unplanned hospital admissions. This is a very interesting subject, as this is a one-time opportunity to analyze such an intervention which has not been performed before. The authors have provided a clear explanation in the introduction of the reason for the intervention and the research questions. It is not clear to me why the authors studied both the probability of an emergency hospital admission and the number of emergency hospital admissions. The methods section starts off with the inclusion of 27,500 patients, but the power calculation is provided at the end. These two should be combined. In the statistics section there is missing an explanation why analyzing the probability of an emergency hospital admission required a binominal log regression model and analyzing the number of emergency hospital admissions required a poisson regression model. The informed reader can probably figure out the difference, but the less well-informed reader probably needs more explanation. The results section is very extended, which provides a clear insight in all the attempts the authors have made to find positive results for the implementation of the named GP-scheme. However they were not able to find any (of relevance). The discussion gives a good insight in the possible reasons for the absence of positive results. My guess is that officially allocating a GP to patients is not necessary: patients have their own preferences and choose to make an appointment with a GP accordingly. Patients that were already accustomed to visiting a specific GP will continue to do so, and others may prefer to visit a GP at a specific date / time, regardless who. These preferences will not change when a specific GP is assigned to the patients. Habits do not change quickly, especially not when top-down implemented by policy makers.
---

	Thank you for submitting this paper, even though it did not provide the results you were looking for. It is relevant to policy makers and should be published, with only minor revisions.
--	---

REVIEWER	Chris van Weel Radboud University Nijmegen, The Netherlands, Department of Primary and Community Care
-----------------	--

REVIEW RETURNED	19-Mar-2019
-------------

GENERAL COMMENTS	The authors have done a great effort to analyse data of organization of care - including the level of 'continuity' and relate this to outcomes in terms of emergency hospital admissions. I applaud their determination, but have to admit that I am unable to interpret their numerical findings in flesh and blood characteristics of the flow of patients through a general practice. What does a BB index of 0.028 or a decrease in BB index of 0.028 mean for Mrs. Jones? Is all continuity lost, or is this a mix-up on a Sunny Friday afternoon in June when one of the partners was away for the day? If I understand the authors, general practices in the UK were forced to tick a box for patients to identify a personal GP. But according to what the authors present on bottom page 6, top page 7, the impact on the actual way of contacting an FP did not change much. Therefore, I would assume, when the measure brought little change, it is unlikely that the measure would have substantial impact - on, for example, emergency admissions. Whatever concerns we have in the UK or the Netherlands, on the current state of family medicine, our systems still provide the highest levels of structured person-centered continuity of care. As a consequence there is little perspective of getting better, and lots of factors will work to make it worse. But how does this reflect to the international readership of the BMJ - FPs in, say, Sri Lanka, Lebanon, Colombo, where they are craving for a health structure that would allow patients to identify with and name a personal FP. Does this paper provide the evidence that this is a spurious ambition? There is a big danger, with the paper presented as it does, that we throw away precious babies, under the assumption that we get rid of the bath water. It remains doubtful that ticking the box of a personal FP would have a radical change in elderly patients' health status, that it would exclude the need of emergency hospital care. Therefore, I do not understand the outcome. I understand the political wishful thinking of reducing hospital care, but from a real life primary care perspective I find this difficult to accept. There might be some ratio in avoiding unwarranted emergency hospital care through better continuity in primary care, but emergency care in total?
---

REVIEWER	Chris Jones Brighton and Sussex Medical School, UK
-----------------	---

REVIEW RETURNED	31-May-2019
-------------

GENERAL COMMENTS	This manuscript presents well defined and thorough analyses of three outcomes, with justifiable conclusions. However, there are
---

	some areas in need of clarification, further justification or simplification: Model specification I think all the details are given regarding specification of the models throughout the manuscript, regarding both fixed and random effects. But it would be good to have this all in one place, showing the whole model for each outcome (possibly once if covariates etc. are all exactly the same for each model). This would also provide clarity on how many levels the models have, as there seems to be clustering within patients and within surgeries. It would also be interesting to see estimates of the ICCs in the model results. Power calculation What software was used for the power calculation? What is the justification for basing this on a reduction of 10% (or 6%)? This power calculation only applies to one of the three outcomes - what about the others? Were any consideration given to clustering in the power calculation? Adjusted models I can't really see the point of presenting the unadjusted model results. They're redundant as the multivariable models provide better estimates - is there a justification for including them? If not, both the text and tables would be simplified by removing them. Model assumptions It needs to be clear that model assumptions were checked and valid. Particularly for Continuity of care index score, the distribution of which is awkward with its floor and ceiling. Figure 1 A more visually attractive way of comparing these two distributions would be with kernel density plots. The current plot makes the score look like it's not continuous. Figure 2 What do the error bars on this plot represent? 95% CIs? Tables The variable labels in a number of the tables should be made more human reader-friendly, particularly in the descriptive statistics table and in the supplementary tables that include the full models Software What statistical software was used for analysis?
--	---

VERSION 1 – AUTHOR RESPONSE

Reviewer: 1

It is not clear to me why the authors studied both the probability of an emergency hospital admission and the number of emergency hospital admissions.

*We have now included a clarification in the section on 'Outcome measures' by adding the following sentence: 'A patient's probability of at least one emergency hospital admission may not necessarily

reflect the number of admissions a patient experienced. The probability might have decreased while the average number of emergency hospital admissions might have increased and vice versa.’ In the statistics section there is missing an explanation why analyzing the probability of an emergency hospital admission required a binominal log regression model and analyzing the number of emergency hospital admissions required a poisson regression model. The informed reader can probably figure out the difference, but the less well-informed reader probably needs more explanation. *We have added in the section on ‘Statistical methods’ that continuity of care was a continuous variable, likelihood of experiencing at least one emergency hospital admission was a binary variable, and number of emergency hospital admissions was a count.

Reviewer 2:

What does a BB index of 0.028 or a decrease in BB index of 0.028 mean for Mrs. Jones? Is all continuity lost, or is this a mix-up on a Sunny Friday afternoon in June when one of the partners was away for the day?

*This is a mean drop in continuity of care index-score. We have avoided applying any value judgement as to whether or not this is “good” or “bad”. We have now added a sentence in the results section: ‘this is a drop in the mean continuity of care by about 6.5%.’ The mean drop of 0.028 is about one tenth of the standard deviation; this is a small downwards shift of the continuity of care (BB index score) distribution. While the change of the mean might only be modest, the distribution of change over time in continuity of care shows a wide variation in change. This distribution is added as a Supplementary Figure 2 to the manuscript and referred to in the section on ‘Outcome: continuity of primary care’. In the discussion section we have now also included the following sentence: ‘The average decrease in continuity of care was small, about 6.5% from baseline, although there was considerable variation across patients and practices.’

If I understand the authors, general practices in the UK were forced to tick a box for patients to identify a personal GP. But according to what the authors present on bottom page 6, top page 7, the impact on the actual way of contacting an FP did not change much. Therefore, I would assume, when the measure brought little change, it is unlikely that the measure would have substantial impact - on, for example, emergency admissions.

*It is certainly possible that most patients were already allocated to a specific GP list, and thus introducing a named-GP policy for older patients might not have changed much in practice. Nevertheless, the policy was introduced with a stated aim of keeping patients out of hospital, and no previous study has evaluated the impact of this policy in terms of achieving that specified objective. The mechanism by which the named GP scheme would lead to reduced admissions was not explicit in the policy, but the implied mechanism was that providing a named GP would lead to improved continuity and better co-ordination of care from one specific doctor, which would in turn lead to reduced admissions. The current analysis therefore still provides useful insights into whether or not the policy has impacted on continuity, as well as whether it has achieved its aims of reducing unplanned hospitalisation. This latter sentence is added in the discussion section of the manuscript. Whatever concerns we have in the UK or the Netherlands, on the current state of family medicine, our systems still provide the highest levels of structured person-centered continuity of care. As a consequence there is little perspective of getting better, and lots of factors will work to make it worse. But how does this reflect to the international readership of the BMJ - FPs in, say, Sri Lanka, Lebanon, Colombo, where they are craving for a health structure that would allow patients to identify with and name a personal FP. Does this paper provide the evidence that this is a spurious ambition? There is a big danger, with the paper presented as it does, that we throw-away precious babies, under the assumption that we get rid of the bath water.

It remains doubtful that ticking the box of a personal FP would have a radical change in elderly patients' health status, that it would exclude the need of emergency hospital care. Therefore, I do not understand the outcome. I understand the political wishful thinking of reducing hospital care, but from a real life primary care perspective I find this difficult to accept. There might be some ratio in avoiding

unwarranted emergency hospital care through better continuity in primary care, but emergency care in total?

*We agree with the reviewer that interpretation of the findings needs to be undertaken with care.

However, we believe the study is justified in the context of strong evidence that poor continuity of care among English patients is associated with an increased risk of experiencing an emergency hospital admission, as well as evidence that continuity of care is dropping in UK practice. We have now included in the discussion section 'Meaning of the study': 'A policy of allocating a named GP in itself is not effective and more sophisticated interventions would be needed to improve continuity of care in the UK or countries with similar healthcare systems. However, it is not possible to tell from our study whether applying an assigned named-GP scheme in a country where continuity of care is not common, might actually lead to improvements in continuity of care and, consequently, hospital use.'

Reviewer 3:

I think all the details are given regarding specification of the models throughout the manuscript, regarding both fixed and random effects. But it would be good to have this all in one place, showing the whole model for each outcome (possibly once if covariates etc. are all exactly the same for each model). This would also provide clarity on how many levels the models have, as there seems to be clustering within patients and within surgeries. It would also be interesting to see estimates of the ICCs in the model results.

*In the section on the results, we have included the heading 'sensitivity analysis' indicating which part of the results are from main or sensitivity analyses. The specification of the models is summarised in the footnotes to Tables 2, 3 and 4. Additional detail on the full model specification is given in Supplementary Tables 2, 3 and 4.

We have now included in the Supplementary Tables 2,3 and 4 the intraclass correlation coefficient (ICC). These tables also show the number of levels in the multilevel analyses.

What software was used for the power calculation? What is the justification for basing this on a reduction of 10% (or 6%)? This power calculation only applies to one of the three outcomes - what about the others? Were any consideration given to clustering in the power calculation?

*We performed the analyses and the power calculation in Stata 15.1 using `sampsi` and `power` commands and included this in the manuscript text. Our study of the impact of continuity of care on emergency hospital admissions suggested that an upward shift across a quartile of the distribution of continuity of care might decrease risk by approximately 10% (Tammes et al. *Annals of Family Medicine* 2017). The power calculation did not adjust for clustering. No prior information was available for the level of an intra-cluster correlation (ICC) for general practices with respect to emergency hospital admission. Since Adams et al. (*Journal of Clinical Epidemiology* 2004) reported a median ICC of 0.01 for a variety of primary care trials, many of which had more proximal outcomes, we have tested the effect on power with an ICC of 0.01, and 0.005. This estimated power as 74% and 90% respectively with the 10% risk reduction we specified.

I can't really see the point of presenting the unadjusted model results. They're redundant as the multivariable models provide better estimates - is there a justification for including them? If not, both the text and tables would be simplified by removing them.

*Reporting of unadjusted results is generally encouraged, as per the STROBE statement (question 16a), providing additional insight into the effect of adjusting for covariates.

It needs to be clear that model assumptions were checked and valid. Particularly for Continuity of care index score, the distribution of which is awkward with its floor and ceiling.

*While the continuity of care index score did have an awkward distribution, our emphasis has been on the change in scores over time, which is much closer to a normal distribution; see Supplementary Figure 2.

A more visually attractive way of comparing these two distributions would be with kernel density plots. The current plot makes the score look like it's not continuous.

*Thank you for the suggestion. However, a kernel density plot fails to emphasize floor and ceiling effects in this variable. We have therefore left Figure 1 as it is.

What do the error bars on this plot represent? 95% CIs?

*Indeed, the error bars represent 95% confidence intervals. This is added in the figure title of Figure 2. The variable labels in a number of the tables should be made more human reader-friendly, particularly in the descriptive statistics table and in the supplementary tables that include the full models

*Following reviewer's suggestion, we have made the variable labels in the descriptive statistics table more human reader friendly.

What statistical software was used for analysis?

*We performed the analysis in Stata 15.1. This is added in the statistical methods section.

VERSION 2 – REVIEW

REVIEWER	Marijke Olthof University Medical Center Groningen, the Netherlands
REVIEW RETURNED	08-Jul-2019

GENERAL COMMENTS	Thank you for your resubmission
---------------------------------

REVIEWER	Chris Jones Brighton and Sussex Medical School, UK
REVIEW RETURNED	10-Jul-2019

GENERAL COMMENTS	No further comments.
----------------------